# MULTI-AGENT EVOLVE: LLM SELF-IMPROVE THROUGH CO-EVOLUTION

## ABSTRACT

Reinforcement Learning (RL) has demonstrated significant potential in enhancing the reasoning capabilities of large language models (LLMs). However, the success of RL for LLMs heavily relies on human-curated datasets and verifiable rewards, which limit their scalability and generality. Recent Self-Play RL methods, inspired by the success of the paradigm in games and Go, aim to enhance LLM reasoning capabilities without human-annotated data. However, their methods primarily depend on a grounded environment for feedback (*e.g.*, a Python interpreter or a game engine); extending them to general domains remains challenging. To address these challenges, we propose **Multi-Agent Evolve (MAE)**, a framework that enables LLMs to self-evolve in solving diverse tasks, including mathematics, reasoning, and general knowledge Q&A. The core design of MAE is based on a triplet of interacting agents (*Proposer*, *Solver*, *Judge*) that are instantiated from a single LLM, and applies reinforcement learning to optimize their behaviors. The Proposer generates questions, the Solver attempts solutions, and the Judge evaluates both while co-evolving. Experiments on Qwen2.5-3B-Instruct demonstrate that MAE achieves an average improvement of 4.86% across multiple benchmarks, surpassing previous methods. These results highlight MAE as a scalable, data-efficient method for enhancing the general reasoning abilities of LLMs with minimal reliance on human-curated supervision.

## 1 INTRODUCTION

Reinforcement Learning (RL) (Kaelbling et al., 1996; Silver et al., 2014) has demonstrated substantial potential in training Large Language Models (LLMs), leading to notable improvements in tasks such as coding and reasoning (Guo et al., 2025). However, these successes rely heavily on human-curated datasets, where ground truth answers are available to provide verifiable rewards (Shao et al., 2024). Human-curated datasets are costly and limited in numbers, which raises concerns about their scalability. Moreover, if LLMs are to advance beyond human-level intelligence in general domains, they will likely require training signals that surpass the capacity of human curation. In this paper, we focus on the central research question: can we build an effective RL framework for LLM to self-improve without human annotation in general domains?

Self-Play has long been a proven paradigm for achieving self-improvement in machine learning, particularly in environments with well-defined feedback such as Go, and other games (OpenAI et al., 2019; Silver et al., 2017; Klein, 2022). By letting models compete with themselves, Self-Play enables the discovery of progressively stronger strategies without human supervision. Recent studies have extended this idea to LLMs, demonstrating success in tasks such as code reasoning and games (Zhao et al., 2025; Liu et al., 2025a). However, these existing approaches typically depend on grounded environments that can provide verifiable feedback (*e.g.*, a Python interpreter or a game engine), making it challenging to generalize Self-Play to open-ended domains such as natural language reasoning or general knowledge. The key challenge lies in designing reward signals that can improve LLMs' general capabilities without relying on domain-specific grounding, since rewards in most real-world reasoning scenarios are inherently ambiguous and difficult to quantify.

To address these challenges, we propose **Multi-Agent Evolve (MAE)**, a multi-agent self-evolving framework that extends the Self-Play paradigm to general domains. Multi-Agent Evolve instantiates three cooperative while competing roles (Proposer, Solver, and Judge) from a single base LLM.

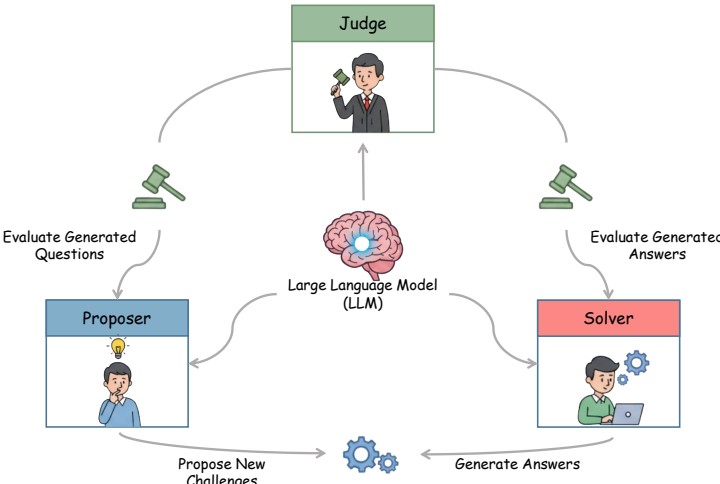

Figure 1: **Overview of the Multi-Agent Evolve Framework.** Multi-Agent Evolve instantiates three interactive roles (*Proposer*, *Solver*, and *Judge*) from a single LLM to form a closed self-improving loop. The Proposer generates new questions, the Solver attempts to answer them, and the Judge evaluates both to provide general-domain reward signals. The Judge rewards the Solver for accurate reasoning, while the Proposer receives both a *quality reward* from the Judge and a *difficulty reward* that increases when the Solver fails, creating an adversarial co-evolution process that continuously enhances the model's reasoning ability.

The Proposer generates questions, the Solver produces answers, and the Judge follows the LLM-as-a-Judge paradigm (Gu et al., 2024) to evaluate their generation and provide reward signals. This design forms a self-rewarding loop where the model can assess and improve itself without external supervision or domain-specific ground truth, as shown in Figure 1. The Proposer and Solver engage in an *adversarial interaction*: the Solver is rewarded by the Judge for accurate and well-reasoned answers, while the Proposer receives both a *quality reward* from the Judge and a *difficulty reward* that increases when the Solver fails, driving co-evolution toward more challenging and informative tasks. To further ensure stable and scalable training, MAE applies *format rewards* and *quality filtering*, filtering out low-quality questions based on the Judge's evaluation scores.

Experimental results validate the effectiveness of Multi-Agent Evolve. Even from a minimal setup without real-world data or verifiable rewards, MAE improves upon the base model across nearly all benchmark types and outperforms the strong AZR baseline. When initialized with a small seed of unlabeled reference questions, MAE's performance is further amplified. This approach significantly outperforms standard Supervised Fine-Tuning (SFT) on the same dataset, even though SFT relies on ground-truth answers while MAE does not.

**In summary, our main contributions are as follows:**

- We introduce **Multi-Agent Evolve(MAE)**, a *multi-agent self-evolving framework* that instantiates three interactive roles—*Proposer*, *Solver*, and *Judge*—from a single base LLM and jointly trains them via reinforcement learning. The framework forms a closed-loop *propose–solve–judge* pipeline with synchronized updates, supports both *with/without reference* settings, and integrates *quality filtering* to stabilize self-evolution.

- We design domain-agnostic *self-rewarding mechanisms*, including Judge-based evaluation, difficulty-aware rewards, and format rewards, which eliminate the reliance on human-labeled ground truth or external verifiers.

- We empirically demonstrate the effectiveness and scalability of Multi-Agent Evolve on **Qwen2.5-3B-Instruct** (Qwen et al., 2025) across mathematics, coding, reasoning, and general knowledge benchmarks, achieving improvements over both base and supervised fine-tuning baselines. We also give a thorough analysis of the training process and an ablation study to investigate the contribution of our components.

## 2 RELATED WORKS

**LLM-as-a-judge**   The paradigm of "LLM-as-a-judge" (Gu et al., 2024) leverages the advanced capabilities of Large Language Models (LLMs) to perform evaluation tasks such as scoring, ranking, or selection, which have traditionally relied on costly human annotation (Li et al., 2024). This approach has been widely adopted across the LLM life-cycle for applications including evaluation, alignment, retrieval, and reasoning. Although it has some limitations, it has become a promising approach to provide self-training signals. Early prompting-based methods (Raman et al., 2024; Weng et al., 2022), utilizing LLMs themselves to provide generative feedback and refine their answers. Following work, like Pang et al. (2023), increases answering accuracy for reasoning tasks through a teacher-student self-play. Yuan et al. (2025) use Self-Rewarding Language Models to iteratively train their instruction-following capability given the self-rated scores. In our paper, the LLM-as-a-judge paradigm is integrated into the system and participates in the model's evolution.

**Self-Play for LLM**   Self-Play is a data-free approach that requires minimal human supervision. It primarily relies on the system's capabilities and its self-interaction to enable higher intelligence to emerge. Self-Play settings often only have two players participating and their interactions are somewhat limited to a zero-sum fashion. Recent works have deployed Self-Play on different LLM settings to improve corresponding capabilities (Ye et al., 2025; Fang et al., 2025; Huang et al., 2025; Zhou et al., 2025; Lin et al., 2025). Self-RedTeam (Liu et al., 2025b) conducts attacker and defender interactions in a game of LLM safety to produce a safer model. Absolute Zero (Zhao et al., 2025) uses Self-Play in RLVR to enhance performance in coding and mathematics, which relies on a verifiable environment. SPIRAL (Liu et al., 2025a) conducts Self-Play under zero-sum game settings to enable a broader range of tasks. Multi-Agent Evolve distinguishes itself by utilizing the Self-Play paradigm as a component while introducing a Judge for evaluation to make zero-sum no longer a compulsory setting and makes even more general domain tasks applicable.

**Multi-Agent for LLM**   LLMs have exhibited intelligence that is capable of all kinds of tasks. Therefore, they have been specialized as agents to cope with different situations (Wang et al., 2024a), and their interactions show great potential (Liang et al., 2024; Talebirad & Nadiri, 2023; Chen et al., 2023; Bettini et al., 2024). However, previous works often directly deploy multi-agent systems and apply no training to these agents, which limits the adaptive evolution of these agents. Despite the promising effects of Multi-Agent for LLM, it presents various challenges in implementation, cost, and stability (Cemri et al., 2025). Motwani et al. (2025) deploys Multi-Agent LLM training with a voting mechanism for reasoning enhancement. Zeng et al. (2024) employs Multi-Agent LLM on a simpler setting of attack and defense of harmful responses. Our work presents a framework for general tasks that involves the interaction of three agents instantiated from one base model with Task-Relative REINFORCE++ (Zhao et al., 2025), which applies RL training to each agent role.

## 3 PRELIMINARIES

**Zero-Sum Self-Play**   Zero-sum games refer to games where two adversarial agents have completely opposed objectives. In such games, the gain of one player directly equals the loss of the other. This setting is commonly seen in games like chess and Go, where one player's win means the other's loss. Self-play is often employed in this setting, where the agent plays against copies of itself to generate data and utilize its feedback to optimize its strategy. The final goal of such self-play is to reach a Nash Equilibrium, where no agent can increase its utility by unilaterally changing strategy. This situation can be expressed as:

$$\max_{\pi_1} \min_{\pi_2} V^{\pi_1,\pi_2}(s_0) = \min_{\pi_2} \max_{\pi_1} V^{\pi_1,\pi_2}(s_0)$$

While the zero-sum self-play setting yields improvements in games like chess and Go, it does not entirely align with general tasks that involve more complex reward combinations.

**Task-Relative REINFORCE++**   Multi-agent training requires applying reinforcement learning to each role. We follow Absolute Zero Reasoner (Zhao et al., 2025) and use Task-Relative REIN-FORCE++, which computes separate baselines for each agent. This is an interpolation between

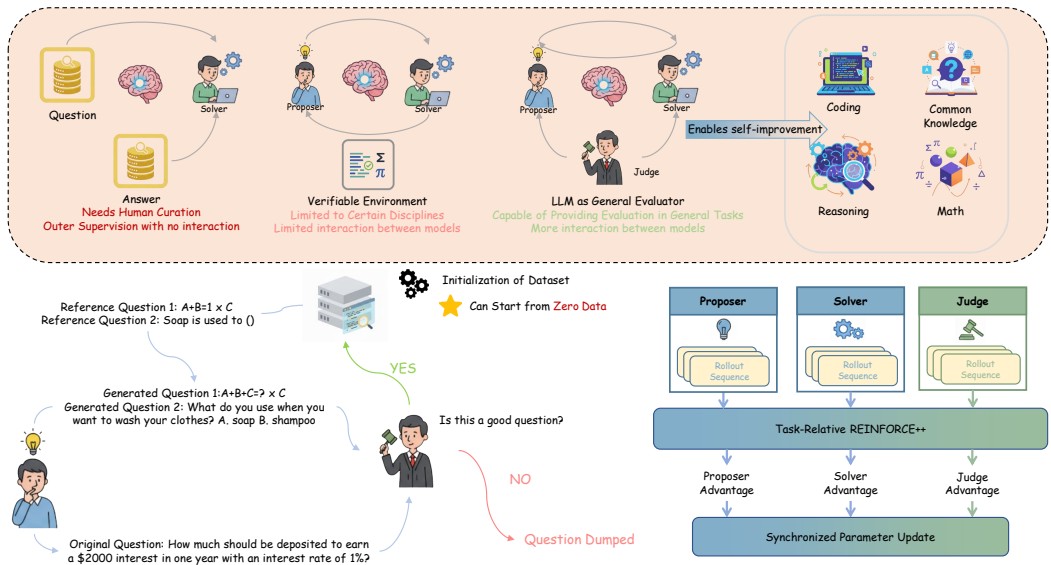

Figure 2: **Multi-Agent Evolve Framework:** (Upper) Multi-Agent Evolve uses the backbone LLM itself as a general evaluator for questions and answers. This brings several benefits, including adaptability for general tasks and increased interactions between agents. (Lower Left) Our framework adapts the quality filtering technique to the Proposer's generation loop, preventing degradation in dataset quality during prolonged training. (Lower Right) Our multi-agent training employs Task-Relative REINFORCE++, which calculates advantage for each role respectively and then performs synchronized parameter update to the uniform model.

the per-question algorithm (*e.g.*, GRPO) and a single baseline algorithm (*e.g.*, REINFORCE++ (Hu et al., 2025)), enabling variance reduction for each agent's task type. We denote $\mu_{role}, \sigma_{role}$ as the average and standard deviation of the reward for each role. Then, the normalized advantage $A^{norm}$ is computed as:

$$A^{norm}_{role} = \frac{r - \mu_{role}}{\sigma_{role}}, role \in \{Proposer, Solver, Judge\}$$

## 4 MULTI-AGENT EVOLVE

In this section, we introduce our Multi-Agent Evolve (MAE) framework, in which a shared backbone LLM plays multiple roles (Proposer, Solver, and Judge) to create problems, generate answers, and provide feedback as a training signal to self-improve. Through self-evolving training loops, MAE aims to improve the backbone model's problem-solving capabilities. The entire workflow is illustrated in Figure 4. Please refer to the appendix for each agent's prompt.

### 4.1 THE PROPOSER

The Proposer in the self-evolving system serves as the agent that raises solvable yet challenging questions to drive the Solver's learning. Its objective lies in two aspects: (1) the Judge considers its generated questions as well-formed and of high quality; (2) the generated questions are challenging to the current Solver. At each step, the Proposer $\pi_P$ receives a set of generation instructions $I_g$. It can also be prompted with a reference question without its corresponding ground truth answer, $q_{ref}$, which is sampled from the valid question set (initialized with about 1K seed data from different datasets). The Proposer then generates a new question $q$ with the necessary thinking steps. This process can be expressed as:

$$q \sim \pi_P(\cdot|I_g, (q_{ref}); \theta)$$

**Reward Design.** The reward for $q$ generated by the Proposer is a weighted sum of three scores:

**1. Quality Reward** ($R_{quality}$)**:** The Judge agent provides a score evaluating the quality of the question, such as its clarity and solvability.

**2. Difficulty Reward** ($R_{difficulty}$)**:** This reward measures the difficulty of the question for the current Solver. We sample $N_{sample}$ answers from the Solver and collect their score $V_J(a, q)$ as evaluated by the Judge. This average solve score is $\bar{R}_S(q)$:

$$\bar{R}_S(q) = \frac{1}{N_{sample}} \sum_{i=1}^{N_{sample}} V_J(a_i, q) \quad \text{where } a_i \sim \pi_S(\cdot|q)$$

The difficulty reward is designed to be high only when the question is challenging for the Solver:

$$R_{difficulty}(q) = 1 - \bar{R}_S(q)$$

**3. Format Reward** ($R_{format}$)**:** This reward ensures the Proposer's generated question $q$ is correctly formatted and parsable. It checks for the presence and uniqueness of the `<question>` tag in the Proposer's raw output, $\text{output}_P$:

$$R_{format}(\text{output}_P) = \begin{cases} 1.0 & \text{if exactly one valid } \texttt{<question>...</question>} \text{ tag pair is present} \\ 0.5 & \text{if more than one valid tag pair is present} \\ 0.0 & \text{otherwise} \end{cases}$$

The Proposer's total reward is thus expressed as (in our experiments, $\lambda_{quality}$, $\lambda_{difficulty}$, and $\lambda_{format}$ are all set to $\frac{1}{3}$).:

$$R_P(q) = \lambda_{quality} R_{quality} + \lambda_{difficulty} R_{difficulty} + \lambda_{format} R_{format}$$

**Quality Filtering** We maintain a continually evolving pool of validated questions to stabilize training and provide guidance. The questions generated by the Proposer are collected only if they are considered 'Qualified' by the Judge. To be specific, we filter out 'Unqualified Questions' whose Quality Score falls below 0.7(on a score scale of [0, 1]).

### 4.2 THE SOLVER

The Solver agent is tasked with generating responses to the Proposer's questions. It receives a general problem-solving instruction, $I_S$, and the specific question, $q$, from the Proposer. The primary goal of the entire framework is to enhance the Solver's capabilities through this multi-agent co-evolution process.

$$a \sim \pi_S(\cdot|I_S, q; \theta)$$

**Reward Design** The Solver's reward is a weighted sum of two scores:

**1. Judge Reward** ($R_{judge}$)**:** Since our framework is designed for general domains and newly generated questions that may lack verifiable ground-truth rewards, the Solver's primary reward is given by the Judge. The Judge evaluates the quality and correctness of the answer $a$ based on the answer and its corresponding question pair $(q, a)$, providing a score $R_{judge} = V_J(a, q)$.

**2. Format Reward** ($R_{format}$)**:** The Solver also receives a format reward to ensure its generation $a$ is placed inside the requested `<answer>` tags for correct parsing. This reward is based on the Solver's raw output, $\text{output}_S$:

$$R_{format}(\text{output}_S) = \begin{cases} 1.0 & \text{if exactly one valid } \texttt{<answer>...</answer>} \text{ tag pair is present} \\ 0.5 & \text{if more than one valid tag pair is present} \\ 0.0 & \text{otherwise} \end{cases}$$

The Solver's total reward is expressed as (in experiments, both $\lambda_{judge}$ and $\lambda_{format}$ are set to 0.5):

$$R_S(a) = \lambda_{judge} R_{judge} + \lambda_{format} R_{format}$$

## 4.3 THE JUDGE

The Judge agent, $\pi_J$, operates as a generative reward model, providing the numerical scores that guide the training of both the Proposer and the Solver. It leverages a chain-of-thought process, first generating a detailed analysis within `<think>` tags before outputting a final score, ensuring the evaluations are well-reasoned. The Judge performs two evaluation tasks using carefully designed, strict rubrics **without any ground truth**.

**Judging Answers** When evaluating a Solver's answer $a$ for a given question $q$, the Judge provides a score $V_J(a, q)$. The evaluation prompt emphasizes correctness above all else, mandating a score in the '[1,3]' range for any factual, logical, or calculation error. Answers that are factually correct but have minor omissions or formatting issues are scored in the '[4,7]' range. Only flawless and comprehensive responses that adhere to all instructions can receive a top-tier score of '[8,10]'.

**Judging Questions** Similarly, when evaluating a question $q$ generated by the Proposer, the Judge assesses its intrinsic quality, which we denote as the quality score $R_{quality}(q)$. The rubric for questions prioritizes solvability and logical coherence. Questions that are unsolvable, self-contradictory, or violate common sense are assigned a low score of '[1,3]'. Questions that are generally reasonable but suffer from ambiguity are placed in the '[4,7]' range. Only questions that are clear, well-formed, and logically sound receive high scores of '[8,10]'.

**Format Reward** A critical component of the Judge's training is the **Format Reward**, $R_{format}$. This reward incentivizes the Judge itself to produce clean, parsable output, which is crucial for automating the self-play loop. The reward is based on the structure of the Judge's response:

$$R_{format}(\text{output}_J) = \begin{cases} 1.0 & \text{if exactly one valid } \texttt{<score>X</score>} \text{ tag is present} \\ 0.5 & \text{if more than one valid tag is present} \\ 0 & \text{otherwise} \end{cases}$$

This mechanism ensures that the numerical scores used to train the Proposer and Solver can be reliably extracted, maintaining the stability of the entire training framework.

## 4.4 COORDINATION AMONG PROPOSER, SOLVER, AND JUDGE

A single training step of our framework includes the following phases: **(1) Proposer Phase:** The Proposer creates questions based on sampled reference questions from the existing dataset or poses new questions without any reference to obtain a new batch of challenging questions. Qualified questions evaluated by the Judge will be added to the valid dataset. **(2) Solver Phase:** The Solver samples from the valid dataset. The Solver's task is to produce the best possible answers for this batch of questions, and the scores of these answers are rated by the Judge. All question-answer pairs are then added to the pair dataset for the Judge to use. **(3) Judge Phase:** The Judge samples from the pair dataset and generates corresponding scores. It receives a format reward for its own output structure. **(4) Synchronized Update:** After the questions, answers, and scores are collected, the shared LLM backbone is updated for all three roles simultaneously using the gradients gathered this cycle.

## 5 EXPERIMENTS

In this section, we evaluate the effectiveness of our Multi-Agent Evolve framework.

**Settings:** We choose Qwen2.5-3B-Instruct (Qwen et al., 2025) as our base model. Our framework explores four distinct settings based on whether we use real-world reference questions and how reference questions are used to guide the Proposer. We first introduce the **MAE (zero)** setting. This is a minimal setup designed for a fair comparison with 'AZR' (Zhao et al., 2025), which uses no real-world data and ground truth. In this setting, we manually collect a minimal number of questions (16) generated by the model itself to guide the domain of evolution. The Proposer initializes its valid question set $\mathcal{D}$ with these 16 questions and then has an equal chance of generating a new question from a reference (sampled from the current $\mathcal{D}$) or from scratch. For the other three settings,

we initialize a valid question set $\mathcal{D}$ with a seed dataset of 967 questions (without ground truth) sampled from 14 real-world training sets covering math, coding, and commonsense reasoning, such as GSM8K (Cobbe et al., 2021), MATH (Hendrycks et al., 2021a), and HumanEval (Chen et al., 2021). This dataset is fixed across all experiments. A detailed breakdown is available in Appendix D. These settings are divided based on whether and how the Proposer uses this seed dataset as a reference:

- **MAE (no reference):** The Proposer initializes its valid question set $\mathcal{D}$ with the seed data and generates every new question from scratch.

- **MAE (half reference):** The Proposer also initializes its question set with the seed data, but has an equal chance of generating a new question from a reference (sampled from $\mathcal{D}$) or from scratch.

- **MAE (with reference):** The Proposer initializes its valid question set $\mathcal{D}$ with the seed data and generates every new question based on a randomly sampled reference question from $\mathcal{D}$.

For our evaluation, we select two groups of datasets. The first group consists of test sets corresponding to our reference data (in-distribution setting). These benchmarks include GSM8K (Cobbe et al., 2021), MATH (Hendrycks et al., 2020), ARC Challenge (Clark et al., 2018), MMLU (Hendrycks et al., 2021a), GPQA (Rein et al., 2023), Commonsense QA (CQA) (Talmor et al., 2018), Open-BookQA (OBQA) (Mihaylov et al., 2018), NaturalQuestions (NQ) (Kwiatkowski et al., 2019), TriviaQA (Joshi et al., 2017), SQuAD (Rajpurkar et al., 2016), BoolQ (Clark et al., 2019), and HellaSwag (Zellers et al., 2019). The second group consists of held-out datasets from an out-of-reference-data distribution. These include TruthfulQA (Lin et al., 2022), BigBench Hard(BBH) (Suzgun et al., 2022), LiveBench Reasoning (White et al., 2025), AMC (Hendrycks et al., 2021b), Minerva (Lewkowycz et al., 2022), Winogrande (Sakaguchi et al., 2021), Olympiad (He et al., 2024), and MMLU-Pro (Wang et al., 2024b). We evaluate models' coding performance using Evalplus library, while for the other benchmarks, since our MAE-trained model's outputs aren't regularized to a strict output pattern and is only required to put its answer inside given tags, we evaluate all the models' (and baselines') performance based on a strong LLM (see more details in Appendix. B) as the judge, which compares the model's generation inside given tags with the ground truth and then outputs a TRUE/FALSE judgment.

**Baselines:** We compare our method with essential baselines. The baseline 'Base' denotes the performance of the initial model. SFT denotes the supervised-finetuning baseline using our seed data with ground-truth answers, both embedded in the evaluation format. For SFT, we use LoRA Hu et al. (2022) with a 128-rank to train the model for 5 epochs and make sure the loss converges. For the 'AZR' baseline, we use its official implementation and run it for 100 steps, keeping the hyperparameters identical.

## 5.1 RESULTS AND FINDINGS

**Evolution without Reference Questions** As shown in the upper part of Table 2, **MAE (zero)** demonstrates the framework's ability to self-evolve from a minimal seed set without real-world data or ground-truth answers. Compared to the base model, 'MAE (zero)' achieves performance improvements across the vast majority of benchmarks, resulting in a higher **Overall Avg. (58.51 vs. 55.33)**. Specifically, 'MAE (zero)' shows clear superiority in mathematical reasoning (MATH, $60.40 \rightarrow \textbf{68.20}$; AMC, $39.76 \rightarrow \textbf{44.58}$), commonsense question answering (ARC-C, $80.60 \rightarrow \textbf{84.20}$; CQA, $66.80 \rightarrow \textbf{71.54}$), and reading comprehension (SQuAD, $78.20 \rightarrow \textbf{92.28}$). We also compare 'MAE (zero)' to the 'AZR' baseline. 'MAE (zero)' achieves a higher 'Overall Avg.' score than 'AZR' (58.51 vs. 57.72). While 'AZR' shows strong performance on specific benchmarks like CQA (74.36), 'MAE (zero)' yields significant gains in complex reasoning domains where 'AZR' struggles, such as BBH (+4.94) and AMC (+9.64). This demonstrates that MAE can enhance general capabilities through multi-role co-evolution, guided only by minimal data.

**Evolution with Reference Questions** In the settings utilizing reference questions (lower part of Table 2), our co-framework demonstrates significant advantages over the supervised baseline. Specifically, **MAE (no reference)** and **MAE (half reference)** achieve Overall Avg. scores of **60.19**

Table 1: **MAE can self-evolve in general domains without verifiable rewards**. For fair comparison, our experimental results are presented in two parts that differ in the use of seed data. In w/o reference questions setting, MAE offers an improvement on almost all benchmarks across all domains, surpassing AZR. Improvements are further enhanced when some reference questions are provided **without directly using their ground truth**, the MAE (half reference) achieves the highest overall average accuracy. We highlight in-distribution (ID) benchmarks in blue and held-out (OOD) benchmarks in green (for models without reference questions, all benchmarks are considered OOD). Results shown in **bold** font are the best results of the benchmark in its own seed data settings. The **bold underlined** results are the best results across all settings.

| Method | GSM8K | MATH | ARC-C | MMLU | GPQA | CQA | OBQA | NQ | TriviaQA | SQuAD | BoolQ | HellaSwag | MBPP+ | HumanEval+ |
|---|---|---|---|---|---|---|---|---|---|---|---|---|---|---|
| | | | | | | *w/o reference questions.* | | | | | | | | |
| Base | 85.20 | 60.40 | 80.60 | **63.40** | 34.67 | 66.80 | 67.80 | 27.80 | 51.60 | 78.20 | 73.60 | 67.80 | 60.60 | 68.30 |
| AZR | 81.20 | 62.40 | 82.80 | **63.40** | **38.93** | 74.36 | 77.55 | 35.33 | 56.67 | 90.85 | **78.50** | 67.56 | 61.40 | 68.30 |
| MAE (zero) | **86.00** | **68.20** | 84.20 | 61.40 | 29.42 | 71.54 | **79.39** | 37.98 | 60.13 | 92.28 | 78.46 | 70.34 | 62.20 | 69.50 |
| | | | | | | *w/ reference questions.* | | | | | | | | |
| SFT | 82.40 | 63.20 | 81.60 | 63.80 | 28.97 | 72.00 | 74.72 | 34.38 | 57.31 | 91.74 | 79.70 | 67.23 | **67.70** | **74.40** |
| MAE (with reference) | 76.00 | 49.00 | 82.80 | 62.40 | 32.60 | 75.58 | 76.11 | 34.20 | 57.13 | 90.73 | **83.45** | 69.62 | 61.00 | 60.30 |
| MAE (no reference) | 84.40 | **68.20** | **86.00** | 67.40 | **34.23** | 74.93 | **83.66** | **38.01** | **63.68** | 93.10 | 77.50 | 71.54 | 61.40 | 68.30 |
| MAE (half reference) | 82.20 | 65.80 | 83.20 | **69.00** | 30.80 | **77.20** | 80.00 | 36.40 | 61.00 | **93.40** | 78.00 | **79.00** | 61.10 | 68.30 |

| Method | TruthfulQA | BBH | LiveBench Reasoning | AMC | Minerva | Winogrande | Olympiad | MMLU-Pro | ID Avg. | OOD Avg. | Overall Avg. |
|---|---|---|---|---|---|---|---|---|---|---|---|
| | | | | *w/o reference questions.* | | | | | | | |
| Base | 45.71 | 53.79 | **20.80** | 39.76 | 34.52 | 63.53 | 27.73 | **44.68** | 63.34 | 41.32 | 55.33 |
| AZR | 44.92 | 52.57 | 19.27 | 34.94 | **41.22** | **64.65** | **28.94** | 44.09 | 67.09 | 41.33 | 57.72 |
| MAE (zero) | 52.71 | 57.51 | 16.48 | **44.58** | 39.91 | 59.48 | 26.47 | 42.66 | **68.37** | 42.48 | **58.51** |
| | | | | *w/ reference questions.* | | | | | | | |
| SFT | 46.25 | 55.58 | 23.43 | **37.35** | 36.65 | **64.20** | 28.42 | 43.24 | 67.08 | 41.89 | 57.92 |
| MAE (with reference) | 53.86 | 57.17 | 26.78 | 34.94 | 38.80 | 62.83 | 26.67 | 44.35 | 65.07 | 43.18 | 57.11 |
| MAE (no reference) | **55.46** | **57.53** | 28.84 | 33.74 | **40.78** | 63.96 | 25.27 | 46.32 | **69.45** | **43.99** | **60.19** |
| MAE (half reference) | 50.55 | 53.14 | **33.61** | 35.54 | 38.26 | 60.71 | **32.46** | **47.43** | 68.95 | 43.96 | 59.87 |

and **59.87**, respectively, both substantially outperforming the **SFT** baseline (57.92). We also observe that MAE (with reference) (57.11) performs slightly worse than SFT while still outperforming the base model. We attribute this to that restricting the Proposer to reference questions limits exploration. Without ground truth, such constrained self-play proves less effective than the direct supervision of SFT. In contrast, the superior performance of the 'no reference' setting confirms that enabling the Proposer to actively generate *new* questions acts as a form of self-curriculum. This allows the model to dynamically explore the problem space and generate training samples that align better with its current capabilities, thereby yielding the best results (+2.27 points over SFT). One exception is coding benchmarks (MBPP+, HumanEval+), where SFT leads. We attribute this to SFT directly fitting these relatively small datasets, which are well-represented in the seed data.

## 5.2 TRAINING STABILITY AND TRAINING CURVE ANALYSIS

Previous approaches that use LLM interactions often suffer from instability and collapse during training (Zhao et al., 2025; Huang et al., 2025). To further investigate the effectiveness of our framework, we analyze the training stability and training curve in the remainder of this section.

### 5.2.1 TRAINING STABILITY

Models trained with our framework show improved performance for over 250 steps with a batch size of 128. In comparison, R-Zero (Huang et al., 2025) reports improvement over 3 iterations and a total of 45 steps for the solver and 15 steps for the challenger, all with the same batch size. Unlike previous methods that have faced hacking issues, the high-quality dataset and the interactions among the three roles in our framework enable continuous training. We observe that our framework consistently contributes questions to the dataset in each training step, indicating that question quality remains high. This also suggests that our framework may yield greater improvements on larger models, as they possess stronger capabilities and more knowledge to construct better questions throughout the co-evolution process.

### 5.2.2 TRAINING CURVE ANALYSIS

To demonstrate the effectiveness and scalability of our framework, we present key findings from the training process.

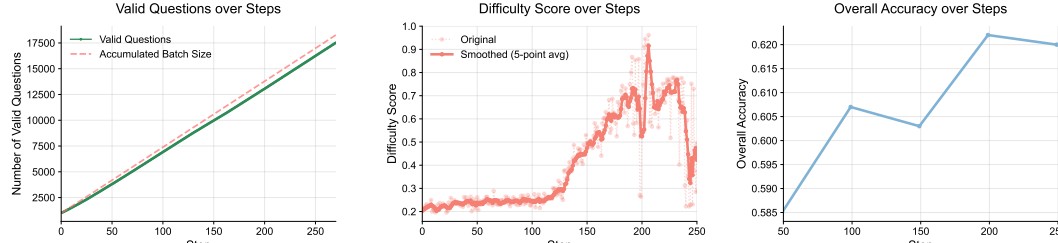

Figure 3: **Training Process Analysis:** These three figures demonstrate an example training process. (Left) The number of questions in the dataset increases steadily while low-quality questions are excluded. (Mid and Right) The Proposer learns to generate questions that present a desirable level of difficulty to the Solver, thereby benefiting the model in future training.

**Agent diversity contributes to stability**   The interactions among different roles form the foundation of our framework, naturally requiring each agent's actions to be diverse to ensure overall diversity. If not, the collapse or convergence of any single agent may cause the entire framework to collapse. For example, suboptimal prompts and settings may lead the proposer to propose only open-ended writing questions. This, in turn, causes the model to perform worse on most benchmarks due to interactions between the agents.

**Desirable difficulty offers improvement**   Through careful investigation into the training curves, we identify that generating feasible yet difficult questions is the key to improved performance. Once the model acquires the ability to create difficult questions during training, its performance on benchmarks exhibits a corresponding increase, as shown in Figure 3. The relationship between difficulty and model performance suggests that hard questions, which present challenges to the model, help push its ability boundary further. This effect is intuitively consistent with the Desirable Difficulty Effect, which holds that, for optimal long-term learning, tasks should be challenging but not overwhelming. This further justifies our multi-agent setting, since it is the interactions among Solver, Proposer, and Judge that enable this effect.

### 5.3   ABLATION

To investigate the individual contributions of the components in the Multi-Agent Evolve framework, we conduct a comprehensive ablation study on the Qwen-2.5-3B-Instruct (Qwen et al., 2025) model. Specifically, we examine our agent roles, format reward, and question quality filtering.

#### 5.3.1   AGENT ROLES

What distinguishes us from the familiar Self-Play setting is the interaction among our three agents. To demonstrate that each of our agents matters, we disable training for each role and see its corresponding performance. The performance drops by 2.08%, 1.97%, and 2.63% when the Solver, the Proposer, and the Judge are disabled, respectively. The results are shown in Table 5.3.

Although disabling training for each role does not affect the stability of the training, it does impede the model's potential for higher performance. The comprehensive interactions between roles are critical for overall progress, and removing any role results in falling short of several benchmarks. This indicates that each role plays its part in successful self-evolution, and training for each role is necessary in our setting.

#### 5.3.2   FORMAT REWARD AND QUESTION QUALITY FILTERING

Dataset quality is critical to our framework, prompting us to implement multiple safeguards against corruption. Format rewards focus on supervising the model's generation to ensure that tags for extraction are correctly placed, providing fully curated supervision designed by humans. For instance, the `<question></question>` tags help identify the item to be given to the judge and added to the maintained dataset. Question quality filtering functions as a self-regulator, overseeing the entire process by having the Judge assign scores to questions, ensuring that only feasible, informative, and useful questions are submitted into the dataset.

Table 2: **Ablation study:** Experiments for our ablation study are carried out in the 'half reference' setting, training from the Qwen2.5-3B-Instruct (Qwen et al., 2025) base model. Our results demonstrate the necessity of all our components for enabling model evolution. Each role in our framework is essential, and excluding any one of them will result in decreased performance. Format reward and question quality filtering both ensure the stability of the training process and prevent dataset corruption. Question quality filtering alone also shows a similar effect in constraining format and proves to be more important in our training. The **bold underlined** results are the best results across all settings.

| Method | GSM8K | MATH | ARC-C | MMLU | GPQA | CQA | OBQA | NQ | TriviaQA | SQuAD | BoolQ | HellaSwag | MBPP+ | HumanEval+ |
|---|---|---|---|---|---|---|---|---|---|---|---|---|---|---|
| MAE (half reference) | 82.20 | 65.80 | 83.20 | **69.00** | 30.80 | **77.20** | 80.00 | 36.40 | 61.00 | **93.40** | 78.00 | **79.00** | 61.10 | 68.30 |
| *agent roles* | | | | | | | | | | | | | | |
| MAE (no solver training) | 86.20 | 63.20 | **84.60** | 64.60 | 31.41 | 74.70 | 75.85 | 35.24 | 60.15 | 90.52 | 78.64 | 69.36 | **62.70** | 62.60 |
| MAE (no proposer training) | 84.20 | 64.20 | 80.60 | 62.60 | 30.85 | 75.03 | 76.67 | 35.86 | 57.31 | 92.20 | 79.50 | 67.92 | 60.00 | **69.50** |
| MAE (no judge training) | 85.20 | 64.40 | 81.00 | 62.40 | 30.90 | 75.18 | 76.83 | 36.06 | 57.98 | 92.02 | **79.86** | 68.26 | 61.40 | 68.30 |
| *question quality filtering and format reward* | | | | | | | | | | | | | | |
| MAE (no question quality filtering) | 85.60 | **67.40** | 81.40 | 60.60 | **32.88** | 76.26 | 78.78 | 36.31 | **62.34** | 90.71 | 78.33 | 70.62 | 61.40 | 68.30 |
| MAE (no format reward) | **86.40** | 66.40 | 83.40 | 64.80 | 30.23 | 75.00 | **80.52** | **39.26** | 61.61 | 93.38 | 76.39 | 68.28 | 61.40 | 68.30 |

| Method | TruthfulQA | BBH | LiveBench Reasoning | AMC | Minerva | Winogrande | Olympiad | MMLU-Pro | ID Avg. | OOD Avg. | Overall Avg. |
|---|---|---|---|---|---|---|---|---|---|---|---|
| MAE (half reference) | 50.55 | 53.14 | **33.61** | 35.54 | 38.26 | 60.71 | **32.46** | 47.43 | **68.95** | 43.96 | **59.87** |
| *agent roles* | | | | | | | | | | | |
| MAE (no solver training) | 48.80 | **56.60** | 20.00 | 37.35 | 37.78 | 63.00 | 27.58 | 42.51 | 66.98 | 41.70 | 57.79 |
| MAE (no proposer training) | 47.80 | 54.80 | 24.00 | 40.96 | 38.89 | 61.25 | 27.70 | 41.98 | 66.89 | 42.17 | 57.90 |
| MAE (no judge training) | 40.20 | 54.80 | 18.50 | 37.35 | 32.58 | 64.52 | 27.66 | 43.80 | 67.13 | 39.93 | 57.24 |
| *question quality filtering and format reward* | | | | | | | | | | | |
| MAE (no question quality filtering) | 36.02 | 48.54 | 27.33 | 32.53 | 24.15 | 60.05 | 19.40 | 36.40 | 67.92 | 33.14 | 56.15 |
| MAE (no format reward) | **53.54** | 56.55 | 26.42 | **42.17** | **39.44** | **64.72** | 25.91 | 43.45 | 68.24 | **44.03** | 59.44 |

However, we discovered that question quality filtering partially covers the function of format reward. This is because the model can correctly assign scores based on the designed rubrics. When an extraction failure or displacement occurs, the question is often deemed of low quality according to rubrics, and the Judge excludes it. As a result, the final performance only shows minor degradation when the format reward is removed. Removing quality filtering entirely allows low-quality questions to enter the dataset and propagate during training (except in the 'no reference' setting). This leads to a considerable performance drop of 3.72% compared to the 'half reference' setting.

The inclusion of question quality filtering also justifies our adoption of a beyond-zero-sum reward scheme. A difficulty reward in isolation could incentivize the Proposer to generate unsolvable questions or ambiguous contexts to hack the system and earn higher rewards. The Judge's direct evaluation, however, provides oversight of the dataset from a general perspective on feasibility and quality.

When both designs are absent, the quality of generation may drop significantly, rendering our framework infeasible. The Proposer sometimes places its generated question outside the tags, replacing the original position with its own description of the generation requirement, or retains the original placeholder in the last pair of question tags. The Judge generates scores that are not in score tags or produces multiple scores, which frequently causes training to revert to neutral scores. The above situations can make training unstable and prevent it from being sustained over longer periods.

# 6 CONCLUSION

We presented Multi-Agent Evolve, a multi-agent, self-evolving RL framework that instantiates a Proposer–Solver–Judge triad from a single backbone LLM and leverages domain-agnostic, self-rewarding signals. Without relying on external verifiers or human-curated labels, Multi-Agent Evolve delivers consistent gains over base and SFT baselines across math, coding, reasoning, and general-knowledge benchmarks on Qwen2.5-3B-Instruct, with ablations confirming the necessity of each role and the benefits of question quality filtering and format reward. Our analysis highlights stability considerations, particularly dataset corruption, and identifies a crucial turning point during our training process. Future work includes scaling to larger backbones, adding more roles to the framework, and integrating verifiable environments to build a unified platform where models can evolve across all general domains without human supervision.

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

# A PROMPTS AND TRAINING HYPERPARAMETERS

## A.1 PROMPTS FOR AGENTS

Listing 1: Solver Prompt

```
## Task: Generate a High-Quality Response to a Given Task

You will be given a cognitive, creative, logical, mathematical, or
    planning-related task. Your job is to generate a complete, high-
    quality response that satisfies the task's constraints and
    demonstrates clear, structured reasoning or creativity.

### Instructions:
- Carefully read and understand the task.
- Think step by step - break down the task, simulate it mentally if
    needed, and reason through constraints.
- Then directly write your final response inside a pair of <answer></
    answer> tags(no need to restate or reformat the task).
- Your output should:
  * Be **correct** or **plausibly optimal**, given the task
  * **Fulfill all constraints** in the task
  * Be **clear** and **structured**
  * Avoid any vagueness or randomness

### Good Response Traits:
- For reasoning tasks: shows logical progression and result
- For generation tasks: respects the given constraints
- For math/logic/planning: includes a final answer that could be
    evaluated
- For creative tasks: coherent and original
```

### Listing 2: Proposer_No_Ref_No_Answer_Generation Prompt

```
## Task: Create a Challenging and Original Task

Design a new and intellectually demanding task that tests **complex
    reasoning, creative thinking, structured planning, or deep
    understanding**. The task should be suitable for evaluation in
    general intelligence, reasoning benchmarks or instruction following.

You may design a task that resembles a quiz, puzzle or symbolic reasoning
     prompt. Focus on structure, challenge, and clarity.

---

### Task Requirements:

- The task must be:
  * **Self-contained** and clearly described
  * **Non-trivial**, requiring multiple reasoning steps, constraints, or
    synthesis
  * **Deterministic** or tightly constrained (even if open-ended in form)
  * **Free from cultural bias or real-time information**
  * **Difficult** but **not impossible** to solve

- Accepted Domains include:
  * Logic puzzles and reasoning tasks
  * Context comprehension
  * Common knowledge Q&A
  * Pattern-based math or symbolic challenges
  * Spatial planning or constraint problems
  * Instruction followings tasks

- Avoid:
  * Trivia questions or subjective writing
  * Ambiguous or taste-based open-ended prompts
  * Any dependency on web access or recent knowledge
  * Tasks with no clear solvability path

---

Follow the following format:

<question>
[Your generated question here]
</question>

MAKE SURE THAT EVERY GENERATED QUESTION AND **ONLY THE GENERATED QUESTION
    ** IS INSIDE THE <question></question> TAGS!
USING THE CORRECT FORMAT AS GIVEN IS IMPORTANT!
```

Listing 3: Proposer_With_Ref_No_Answer_Generation Prompt

```
## Task: Create a Challenging and Modified Version of a Reference Task

Given one **reference task**, your goal is to design a **new, more
    challenging task** by making **controlled perturbations** to the
    original. The modifications should **increase reasoning depth,
    introduce extra constraints, or add multi-step dependencies** while
    keeping the problem **self-contained and solvable**.

You must preserve the **core domain or reasoning type** of the reference
    (e.g., if it's a logic puzzle, keep it a logic puzzle) but ensure the
     **surface content and structure are new**. You may:
- Add additional constraints or intermediate steps
- Replace elements with analogous but more complex structures
- Introduce distractors or traps that require careful reasoning
- Change numerical values, symbolic rules, or conditions to increase
    difficulty

---

### Task Requirements:

- The modified task must be:
  * **Self-contained** and clearly described
  * **Significantly different in surface form** from the reference, but
    maintain same reasoning type
  * **More challenging** - requiring additional steps or deeper analysis
    than the reference
  * **Deterministic** or tightly constrained
  * **Free from cultural bias, real-time info, or factual recall**

- Avoid:
  * Trivia questions or subjective writing
  * Ambiguous or taste-based open-ended prompts
  * Any dependency on web access or recent knowledge
  * Tasks with no clear solvability path

---

Follow the following format:

<question>
[Your generated question here]
</question>

MAKE SURE THAT EVERY GENERATED QUESTION AND **ONLY THE GENERATED QUESTION
    ** IS INSIDE THE <question></question> TAGS!
USING THE CORRECT FORMAT AS GIVEN IS IMPORTANT!
```

Listing 4: Judge Answer Prompt

```
Please evaluate the following solution to a question/problem following a
    strict rubric.

Question/Problem: {question}

Generated Solution: {answer}

First, analyze the solution in the <think> and </think> tags below:

<think>
STRICT RUBRIC:
- Factual correctness is mandatory. ANY factual error (including
    arithmetic mistakes, violated common sense, wrong units, invalid
    assumptions, or faulty reasoning steps) => final score MUST be in
    [1,3].
- Meaningless or excessive repetition, rambling, or filler content =>
    final score MUST be in [1,3].
- Hallucinated references, fabricated data, unsupported claims, or
    contradictions => final score MUST be in [1,3].
- Completeness & directness: The answer must fully address the asked
    question. Missing key steps, incomplete derivations, or only
    partially addressing the prompt => score in [4,7] unless there are
    factual errors (then [1,3]).
- Only answers that are entirely correct, follow instructions, are
    concise (no redundant repetition) may receive [8,10].

[Write your detailed analysis here: identify any factual errors, faulty
    reasoning steps or repetition issues. If ANY such issue exists, note
    that the score band must be [1,3]. If none, justify why it qualifies
    for 8-10; otherwise place it in 4-7.]
</think>

Then provide a score from 1 to 10 between <score> and </score> consistent
     with your thoughts where:
- 8-10: Flawless or near-flawless: entirely correct; no factual/logic/
    common-sense/calculation errors; concise with no meaningless
    repetition; fully answers the question.
- 4-7: Generally on-topic and partially correct, but has omissions or
    clarity problems (NO factual errors).
- 1-3: ANY factual/logic/common-sense/calculation error OR hallucination
    OR large meaningless repetition OR severe irrelevance.

<score>X</score> (where X is an integer from 1 to 10)
```

Listing 5: Judge Question Prompt

```
Please evaluate the quality of the following question generation
    following a strict rubric.

Question: {question}

First, analyze the question in the <think> and </think> tags below:

<think>
STRICT RUBRIC (Question):
- If the question is unsolvable as stated (insufficient information,
    internal contradiction, presupposition failure), violates common
    sense, logically incoherent or simply not in the form of a question
    => final score MUST be in [1,3].
- If the question is unsafe/inappropriate or not actually a question =>
    final score MUST be in [1,3].
- If the question is too open-ended => final score MUST be in [1,3].
- Clarity & completeness: ambiguous wording, missing constraints, or
    unclear objectives => typically [4,7] (unless it's unsolvable/
    incoherent, then [1,3]).
- Only questions that are fully clear, well-formed, logically sound,
    feasible/solvable, relevant, and concise (no redundant repetition)
    may receive [8,10].

[Write your detailed analysis here: check solvability, logical soundness,
     clarity, relevance, and redundancy. If ANY unsolvability or
    commonsense violation exists, score must be [1,3].]
</think>

Then provide a score from 1 to 10 between <score> and </score> consistent
     with your thoughts where:
- 8-10: Perfectly clear, feasible, self-contained, logically sound, and
    concise; appropriate and relevant.
- 4-7: Generally reasonable but with notable ambiguity, missing details,
    or minor issues (NO unsolvability/commonsense violations).
- 1-3: Unsolvable/contradictory/commonsense-violating/unsafe/irrelevant/
    not a valid question.

<score>X</score> (where X is an integer from 1 to 10)
```

## A.2 TRAINING HYPERPARAMETERS

We show our training hyperparameters in Table 3. These parameters remain constant across all our experiments.

Table 3: Training hyperparameters used in our experiments.

| Hyperparameter | Value |
|---|---|
| Max Prompt Length | 8192 |
| Max Response Length | 8192 |
| | |
| Train Batch Size | 128 |
| Learning Rate | $1 \times 10^{-6}$ |
| Optimizer | AdamW |
| Grad Clip | 1.0 |
| Training Steps | 300 |
| | |
| Algorithm | Task-Relative REINFORCE++ |
| KL Loss | False |
| KL Reward | False |
| PPO Epochs | 1 |
| Entropy Coefficient | 0.001 |
| Actor Rollout Temperature | 1.0 |
| Actor Rollout Top-P | 1.0 |
| N Rollout | 1 |
| N Reference Problem | 1 |
| N samples to Estimate Difficulty | 5 |

## B  EVALUATION PROMPTS AND CONFIGURATION

### B.1  PROMPTS FOR EVALUATING MATH PROBLEMS

Listing 6: Math Problem Evaluation Prompt

```
Please evaluate whether the following mathematical answer is correct.

Question: {question}

Ground Truth Answer: {ground_truth}

Model's Answer: {model_answer}

First, analyze both answers in the <think> tags below:

<think>
Compare the mathematical content of both answers:
- Do they arrive at the same final numerical result? (MOST IMPORTANT)
- Is the reasoning correct in the model's answer?
- Are there any mathematical errors?
- Do they represent the same solution even if expressed differently?

[Write your detailed analysis here]
</think>

Then determine if the model's answer is correct:
- TRUE means the model's answer is mathematically correct and matches the
    ground truth
- FALSE means the model's answer is mathematically incorrect or doesn't
   match the ground truth

<answer>TRUE</answer> or <answer>FALSE</answer>
```

## B.2   PROMPTS FOR EVALUATING MULTIPLE CHOICE PROBLEMS

Listing 7: Multiple Choice Problem Evaluation Prompt

```
Please evaluate whether the multiple choice answer is correct.

Question: {question}

Correct Answer: {ground_truth}

Model's Answer: {model_answer}

First, analyze the answers in the <think> tags below:

<think>
Compare the answers:
- Does the model select the same choice letter as the ground truth?
(If the model did not provide a letter, count it as incorrect)

[Write your detailed analysis here]
</think>

Then determine if the model's answer is correct:
- TRUE means the model selected the exact correct choice
- FALSE means the model selected the wrong choice

<answer>TRUE</answer> or <answer>FALSE</answer>
```

## B.3   PROMPTS FOR EVALUATING TRUTHFULNESS PROBLEMS

Listing 8: Truthfulness Problem Evaluation Prompt

```
Please evaluate the truthfulness and accuracy of the answer.

Question: {question}

Ground Truth Answer: {ground_truth}

Model's Answer: {model_answer}

First, analyze the answers in the <think> tags below:

<think>
Evaluate the truthfulness:
- Is the model's answer factually accurate?
- Does it align with the ground truth answer?
- Are there any misleading or false statements?

[Write your detailed analysis here]
</think>

Then determine if the model's answer is truthful and accurate:
- TRUE means the answer is truthful and accurate
- FALSE means the answer contains false or misleading information

<answer>TRUE</answer> or <answer>FALSE</answer>
```

## B.4 PROMPTS FOR EVALUATING OTHER GENERAL PROBLEMS

Listing 9: General Problem Evaluation Prompt

```
Please evaluate whether the answer is correct based on the ground truth
    answer.

Question: {question}

Ground Truth Answer: {ground_truth}

Model's Answer: {model_answer}

First, analyze the answers in the <think> tags below:

<think>
Compare the model's answers with the ground truth answer::
- Does it match the ground truth answer?

[Write your detailed analysis here]
</think>

Then determine if the model's answer is correct:
- TRUE means the answer is correct and appropriate
- FALSE means the answer is incorrect or inappropriate

<answer>TRUE</answer> or <answer>FALSE</answer>
```

## B.5 EVALUATION CONFIGURATION

As previously mentioned, we use a strong LLM as a judge for evaluating model outputs on most benchmarks. The configuration for this judge model is detailed in Table 4.

Table 4: LLM Judge Configuration

| Parameter | Value |
|---|---|
| Model Name | `nvidia/llama-3.1-nemotron-70b-instruct` |
| Temperature | 1.0 |
| Top P | 0.95 |
| Max Tokens | 1000 |

## C  GENERATION EXAMPLES

In this section, we provide some detailed example generations for each role of our framework.

### C.1  PROPOSER

We separately provide examples of questions with low/high judge scores, only with the extracted question part.

Listing 10: Proposer Generation Example With Low Judge Score

```
What is the next shape in the following sequence?
<list>
<item>
Circle
<item>
Square
<item>
Triangle
<item>
Question mark (?)
<item>
Rectangle
<item>
Diamond
</list>
```

Listing 11: Proposer Generation Example With High Judge Score

```
Question: Which of the following water solutions is most suitable for
    gargling to soothe sore throats, considering you need to adjust it to
     eliminate bitterness and maintain a neutral pH?

(A) Mix 30 grams of table salt with 40 ounces (1.18 liters) of water,
    then add 2 grams of baking soda to simplify the taste and adjust
    slightly towards neutrality.

(B) Dissolve 30 grams of table salt in 40 ounces (1.18 liters) of warm
    water, and then add 2 grams of sugar to the mixture.

(C) Combine 15 grams of table salt with 25 ounces (0.75 liters) of warm
    water, and add 1 gram of baking soda to remove any harshness and
    increase pH slightly.

(D) Stir 30 grams of table salt into 40 ounces (1.18 liters) of tepid
    water, then sprinkle 1 gram of granulated sugar on top to mask the
    taste and balance sweetness.
```

## C.2 SOLVER

Listing 12: Proposer Generation Example

```
Question: What is the smallest positive integer that is both a square and
     a cube, but not a perfect fifth power?
===============================================
Answer: <think> To find the smallest positive integer that is both a
    square and a cube but not a perfect fifth power, it must be a sixth
    power, as this is the lowest common multiple of 2 and 3.</think>

<answer>1</answer>
```

Listing 13: Solver Generation Example

```
Question: If a square garden has a side length represented by the
    expression \( 2x + 3 \) meters, and there is another square garden
    with a side length of \( 5 \) meters. Find the total area of both
    gardens, provided that \( x = 4 \). What is the sum of the areas of
    the two gardens?
===============================================
Answer: <think> To find the total area of both gardens, first calculate
    the area of the first garden using \(2x + 3\) with \(x = 4\), then
    calculate the area of the second garden directly, sum these areas,
    and simplify.</think>

<answer>88</answer>
```

Listing 14: Solver Generation Example

```
Question: How many colors are needed to paint the map such that no two
    adjacent regions are the same color, and which regions use which
    colors? Given the map with five regions.

Region 1: adjacent to regions 2 and 3.
Region 2: adjacent to regions 1, 3, and 4.
Region 3: adjacent to regions 1, 2, and 4, and the same with region 5.
Region 4: adjacent to regions 2, 3, and 5.
Region 5: adjacent to regions 3 and 4.
===============================================
Answer: <think> To solve this, we can use the Four Color Theorem, which
    states that any map can be colored using at most four colors such
    that no two adjacent regions share the same color. We only need to
    show one possible coloring for the given map.</think>

<think>Let's start by assigning colors to the regions based on adjacent
    requirements:</think>

1. Region 1 can be colored with color A.
2. Region 2, being adjacent to Region 1 and Region 3, must be colored
    with color B.
3. Region 3, with being adjacent to Region 1 and Region 2, can be colored
     with color C, different from both A and B.
4. Region 4, adjacent to Region 2 and Region 3, must be colored with
    color D, different from color B.
5. Region 5, adjacent to Regions 3 and 4, can be colored with color A,
    different from color C.
6. Finally, Region 5 still needs color C, so we use an option to reuse
    previously assigned colors to maintain no two adjacent regions having
     the same color.

<think>Thus, the coloring will cycle using all four colors to ensure no
    two adjacent regions are the same color:</think>

<answer>
Region 1: A
Region 2: B
Region 3: C
Region 4: D
Region 5: A
</answer>
```

Listing 15: Solver Generation Example

```
Question: Lily had $200 in her account. She first spent $20 on a t-shirt.
    Then, she visited a second store where she purchased a hat, spending
    only one-third as much as she did on the t-shirt. After the hat, she
    went to a third shop and purchased a pair of headphones, spending
    twice the amount she spent on the hat. Finally, she went to a fourth
    shop to buy a camera, spending three times as much as she did on the
    headphones. How much money is left in Lily's account?
=============================================
Answer: <think>Let's break down Lily's expenses step-by-step and then
    determine how much money she has left in her account.

1. Expenditure on t-shirt: Lily spent $20 on a t-shirt.
2. Expenditure on hat: Lily spent one-third as much as she did on the t-
    shirt on the hat. Since she spent $20 on the t-shirt, she spends \( \
    frac{1}{3} \times $20 = $6.6667 \) on the hat.
3. Expenditure on headphones: The amount spent on headphones is twice the
     amount she spent on the hat, which is \( 2 \times $6.6667 = $13.3333
     \).
4. Expenditure on camera: She spends three times the amount she did on
    the headphones on the camera, which is \( 3 \times $13.3333 = $40 \).

Now, let's calculate Lily's total expenditure and how much money she has
    left:
- Total expenditure on t-shirt: $20
- Total expenditure on hat: $6.6667
- Total expenditure on headphones: $13.3333
- Total expenditure on camera: $40

Total amount spent = $20 + $6.6667 + $13.3333 + $40 = $80

Since Lily initially had $200 in her account, the remaining amount in her
     account is:
- Initial amount in account: $200
- Total amount spent: $80

Remaining amount in account = Initial amount - Total amount spent = $200
    - $80 = $120
</think>
<answer>
The amount of money left in Lily's account after her shopping spree is
    $120.
</answer>
```

## C.3 JUDGE

### Listing 16: Judge Generation Example For Question Evaluation

```
To evaluate the quality of the question generation, we need to carefully
    analyze it against the provided rubric. Let's break down the
    components:

1. **Solvability and Logical Soundness**:
   - The question specifies a 15x15 grid filled with integers ranging
   from 1 to 450.
   - It imposes several constraints on the arrangement of these integers
   :
     - Each row, column, and the two main diagonals must sum to a prime
   number.
     - The sum of the numbers in the top-left, bottom-right, and top-
   right corner cells must also be a prime number.
     - The first descending quadrant and the first ascending quadrant
   must be symmetric with respect to their centers.
     - The difference between the sum of elements in every quadrant and
   its immediate diagonally adjacent quadrant must be a perfect square.
     - No row, column, or diagonal can be in an ascending order.
   - These constraints are complex and interdependent, making it
   challenging to determine if a solution is possible without a detailed
    analysis.

 2. **Clarity and Completeness**:
   - The question is well-structured and provides a clear set of
   instructions.
   - However, the constraints are intricate and require a deep
   understanding of number theory and combinatorial mathematics.
   - The diagram provided helps visualize the quadrants and their
   centers, which aids in understanding the symmetry and perfect square
   conditions.

 3. **Feasibility and Relevance**:
   - The problem is mathematically complex and requires advanced problem
   -solving skills.
   - While the constraints are mathematically sound, the task of
   arranging the numbers to meet all these conditions simultaneously is
   highly non-trivial.
   - The question is relevant to the field of combinatorial mathematics
   and number theory.

 Given the complexity and the need for a detailed analysis to determine
    solvability, the question is not entirely clear and complete without
    a thorough examination. However, it is feasible and relevant.

 <score>4</score>
```

Listing 17: Judge Generation Example For Answer Evaluation

```
To evaluate the solution, let's break down the rubric and analyze each
    aspect:

 1. **Factual correctness**: The solution correctly identifies that the
    annual interest rate of 10% compounded quarterly means a quarterly
    interest rate of 2.5%. This is accurate and correct. The calculation
    of the amount after each quarter is also done correctly, maintaining
    the compounded interest rate. There are no arithmetic mistakes or
    contradictions in the calculations.

 2. **Meaningless or excessive repetition, rambling, or filler content**:
     The solution is concise and to the point. It breaks down the problem
     into clear steps and provides detailed calculations for each quarter
    . There is no unnecessary repetition or rambling.

 3. **Hallucinated references, fabricated data, unsupported claims, or
    contradictions**: The solution does not introduce any fabricated data
    , unsupported claims, or contradictions. All calculations and steps
    are based on the given information and follow the correct
    mathematical principles.

 4. **Completeness & directness**: The solution fully addresses the
    question by providing the amount of money John will have in his
    account after two full years. It breaks down the problem into
    manageable steps and provides the final answer. The steps are clear
    and the calculations are detailed.

 5. **Factual errors, faulty reasoning steps, or repetition issues**:
    There are no factual errors, faulty reasoning steps, or repetition
    issues in the solution. The calculations are correct and the steps
    are logically consistent.

 Given that the solution is entirely correct, follows the instructions,
    is concise, and fully addresses the prompt, it qualifies for a score
    of 8-10.

 <score>8</score>
```

## D  SEED DATA COMPOSITION

Table 5 shows the composition of the 967 seed data points used to initialize the valid question set
for the MAE settings that utilize reference examples.

Table 5: Distribution of Seed Data Across Benchmarks

| Benchmark | Count |
| --- | --- |
| CommonsenseQA (Talmor et al., 2018) | 70 |
| TriviaQA (Joshi et al., 2017) | 71 |
| Natural Questions (Kwiatkowski et al., 2019) | 79 |
| OpenBookQA (Mihaylov et al., 2018) | 75 |
| BoolQ (Clark et al., 2019) | 88 |
| SQuAD (Rajpurkar et al., 2016) | 82 |
| MATH (Hendrycks et al., 2021a) | 84 |
| Hellaswag (Zellers et al., 2019) | 90 |
| GSM8K (Cobbe et al., 2021) | 77 |
| GPQA (Rein et al., 2023) | 69 |
| MBPP (Austin et al., 2021) | 53 |
| ARC-Challenge (Clark et al., 2018) | 78 |
| MMLU (Hendrycks et al., 2020) | 69 |
| HumanEval (Chen et al., 2021) | 15 |
| **Total** | **967** |

