# OpenReview forum: "Multi-Agent Evolve: LLM Self-Improve through Multi-Agent Co-evolution"
_ICLR.cc/2026/Conference — Submitted to ICLR 2026_

### Official Review · Reviewer_Pegt · 2025-10-31

**Soundness:** 3
**Presentation:** 3
**Contribution:** 3
**Rating:** 2
**Confidence:** 4

**Summary:**

The paper proposes Multi-Agent Evolve, a self-improvement framework for large language models (LLMs) using a tri-agent co-evolution setup: Proposer, Solver, and Judge, all instantiated from a single base LLM. The framework uses reinforcement learning with self-generated rewards to improve model performance across reasoning, math, coding, and knowledge tasks without requiring human-annotated data. The authors report improvements over baselines like supervised fine-tuning (SFT) on Qwen2.5-3B-Instruct, with ablations showing the importance of each agent and design choice.

**Strengths:**

* The idea of using three distinct roles (Proposer, Solver, Judge) co-evolving via RL in a self-play-like setup is novel and extends prior self-play methods to more general domains beyond zero-sum games or verifiable environments.

* The paper addresses the problem of human-curated reward dependence in LLM training, presenting a scalable and domain-agnostic alternative for general-purpose use.

**Weaknesses:**

* Reward Design is Under-specified: The exact reward functions for the Solver and Judge are not clearly defined. For instance, it is unclear whether the Solver is directly optimized using $V_J(a_i, q)$, and whether the Judge is only trained with the format reward or also receives other signals. This ambiguity hampers reproducibility and understanding.

* Experimental Results Lack Consistency and Explanation: While some benchmarks show improvement, the results are mixed across datasets. For example, MAE(with-ref) performs worse than MAE(no-ref), which is counterintuitive and unexplained. The paper does not provide a convincing hypothesis or analysis for these inconsistencies.

**Questions:**

1. Clarification of Reward Functions:

   * What is the exact reward function used for the Solver? Is it simply $V_J(a_i, q)$?

   * What is the complete reward for the Judge? Is it only the format reward, or does it include other signals?

2. Definition of Symbols: Please define $\mu_{role}$ and $\sigma_{role}$ in the Task-Relative REINFORCE++ formula.

3. Explanation of Mixed Results:

   * Why does MAE(with-ref) underperform MAE(no-ref) and MAE(half-ref)? This is counterintuitive, as reference data is expected to help. Could you provide a hypothesis or analysis?

   * Why do some datasets (e.g., GSM8K, MATH) show consistent gains while others (e.g., HumanEval+, TruthfulQA) show mixed trends?

4. Computational Cost: How much GPU resource and training time are required for the full training pipeline? Is the method practical for larger models?

5. Formatting and Typographical Issues:

* The abstract is missing a numerical value: "an average improvement of %"

* Please check the quotation marks carefully: for example, "answer-then-refine" in the last line of the first paragraph of Section 4.1, "Unqualified Questions" in the Question Collection paragraph, and the Judging Answers paragraph in Section 4.3.

* Figure 2 caption refers to "Left/Right" but should be "Top/Bottom".

---

> ### Author Response · Authors · 2025-11-25
>
> **[Reward Settings]** Thank you for raising this question. The revised manuscript has given an explicit expression for each role reward (**Section 4**). The solver only receives the Judge score as the reward.
>
> Regarding the Judge's reward issue, the judge only receives the format reward. Our initial motivation is self-evolution without any supervision, including ground truth. Under this setting, we have found no proper reward for the Judge. It is possible to provide supervision for the Judge by aligning it with curated data of different quality during training. However, this is out of the scope of our paper and therefore not implemented.
>
> While there is no explicit expression of the Judge’s objective and reward, the Judge’s task actually consists of deciding the quality of questions and answers. These two subtasks correspond to the model’s abilities as Proposer and Solver. **A better ability to propose and solve tasks leads to a more accurate judgment of question/answer quality**. The evolution of the Proposer and Solver directly contributes to the Judge's evolution, enabling the co-evolution of the entire system.
>
> **[Half Reference Outperforms]** We have clearly stated our three settings in our original manuscript. The “half reference” and “no reference” settings intuitively have an advantage, as they can generate original questions that are beneficial for model improvement, rather than relying solely on reference questions and their variations. Experiment results also validate this intuition. Nonetheless, we have provided a more precise explanation of our settings (**Section 5**) for you.
>
> **[Mixed Results]** There is no guarantee that improvements can be achieved on every benchmark. There are also too many factors that may influence performance on a single benchmark of a model checkpoint. However, MAE achieves the best overall performance through repeated experiments.
>
> **[Writing Issues]** We have corrected the typo in the revised manuscript. All formatting and typographical issues have been addressed. Thank you for acknowledging us on these issues.
>
> **[Resource Usage and Scalability]** Training is done on 8xH20 for 12 hours and can be applied to larger models. Please refer to the general official comment and our revised manuscript for more results.

---

### Official Review · Reviewer_dmfX · 2025-10-31

**Soundness:** 2
**Presentation:** 2
**Contribution:** 2
**Rating:** 2
**Confidence:** 3

**Summary:**

The paper introduces a framework for training multi-agent systems through self-play. The system consists of three LLMs: a proposer, which generates questions; a solver, which attempts to answer them; and a judge, which evaluates both the questions and the corresponding answers. Each component receives rewards that can be leveraged in reinforcement learning to improve the agents’ performance. The approach is implemented using Qwen-2.5 3B and evaluated on multiple datasets.

**Strengths:**

1. Quantitative improvements across several datasets indicate that the method is promising in terms of training effectiveness.
2. The manuscript is clearly written and easy to follow.

**Weaknesses:**

I identify several issues with the current version of the manuscript:

1. The main concern is that the framework is evaluated on a single backbone LLM of relatively small size (3B parameters). The observed improvements might simply result from the limited capabilities of Qwen-2.5 3B, which can be enhanced even with noisy training data. How does the method perform with other architectures or larger models? Although it is always possible to include new backbones, I think that for contributions such as this one at least one model with 7-8B parameters is necessary.
2. The novelty of the contribution appears limited. The paper closely resembles MALT (appropriately cited in the manuscript), with the distinction that MALT uses ground-truth data instead of an LLM-as-a-judge and applies DPO for post-training. Architecturally, the contributions are similar, with slightly different roles for the agents. While removing the reliance on ground truth using a judge is an important difference, I remain unconvinced about its necessity (see below).
3. If seed questions are required to bootstrap training, the motivation for replacing the ground truth with a judge seems weak. For such questions, ground-truth answers are typically available, especially in math-oriented tasks, where correct solutions can be computed once the problem is formalized. Why not use them? In the proposed experiments, there is no clear use case demonstrating an advantage of employing a judge over ground truth.

As a minor comment, since no ground truth is used, the method’s performance is inherently limited by that of the judge. This likely explains the use of a much larger network (e.g., a LLaMA 70B variant) to compensate for the shortcomings of Qwen-2.5 3B. What happens if the judge underperforms?

**Questions:**

1. What happens with other models, and in absence of experiments, why evaluation on Qwen 2.5 3B should be sufficient?
2. Can authors clarify on novelty with respect to MALT?
3. Which is one case in which the judge is better than ground truth, and we do not realistically have ground truth?
4. Are there any collapse due to the judge performance?

---

> ### Author Response · Authors · 2025-11-25
>
> **[Bigger Models]** We understand the reviewer’s concern regarding the diversity of base models and the scalability of our framework.
>
> It is worth noting that RL-based self-play methods are computationally intensive due to the multi-agent interactions (Proposer, Solver, Judge) and extensive sampling required. **Recent relevant works such as Kuba et al. (2025, *Language self-play for data-free training*) also primarily validate their self-play methods on 3B models.**
>
> **However, to address your concern and demonstrate scalability, we conducted additional experiments on larger and newer models: Qwen3-4B-Instruct-2507 and Qwen2.5-7B-Instruct.** (Due to resource limitations, we were unable to train for long enough or to conduct repeated experiments for each model.)
>
> Comparing our method (MAE) against the Base model and SFT baseline, we observed consistent improvements:
>
> - On **Qwen3-4B-Instruct-2507**: MAE (half reference) outperforms the Base model by **0.48** points and SFT by **0.83** points on the overall average. Notably, MAE (half reference) demonstrates significant generalization capability, surpassing the Base model by **2.67** points on OOD benchmarks.
> - On **Qwen2.5-7B-Instruct**: MAE (half reference) achieves a clear lead over both Base (**+0.66**) and SFT (**+0.77**), proving that our multi-agent co-evolution framework generalizes effectively to larger model scales.
>
> Detailed results are provided in the table below.
> | Model | Method | GSM8K | MATH | ARC-C | MMLU | GPQA | CQA | OBQA | NQ | TriviaQA | SQuAD | BoolQ | HellaSwag | MBPP+ | HumanEval+ | TruthfulQA | BBH | LiveBench | AMC | Minerva | Winogrande | Olympiad | MMLU-Pro | ID Avg. | OOD Avg. | Overall Avg. |
> | --- | --- | --- | --- | --- | --- | --- | --- | --- | --- | --- | --- | --- | --- | --- | --- | --- | --- | --- | --- | --- | --- | --- | --- | --- | --- | --- |
> | **Qwen2.5-7B** | Base | 91.40 | 74.40 | 91.00 | **74.40** | 30.27 | 81.56 | 87.83 | 42.16 | 67.15 | 96.20 | **87.98** | 81.06 | **68.00** | 74.40 | 63.19 | 69.57 | 27.28 | 49.40 | 53.35 | **68.16** | 37.93 | 55.45 | 74.84 | 53.04 | 66.92 |
> |  | SFT | **92.00** | 74.00 | 91.00 | 74.20 | 30.65 | 81.56 | 88.00 | 41.86 | **67.83** | 95.70 | 87.49 | 81.06 | 67.70 | 74.40 | 63.97 | 70.00 | 26.61 | **50.60** | 49.02 | 68.08 | **38.13** | 56.03 | 74.82 | 52.81 | 66.81 |
> |  | MAE (half reference) | 91.80 | **76.20** | **93.40** | 73.20 | **34.42** | **82.85** | **91.34** | **45.60** | 67.80 | **96.52** | 87.83 | **81.39** | 65.30 | **76.20** | **67.07** | **71.09** | **28.92** | 42.17 | **57.15** | 67.13 | 32.69 | **56.79** | **75.99** | **52.88** | **67.58** |
> |  |  |  |  |  |  |  |  |  |  |  |  |  |  |  |  |  |  |  |  |  |  |  |  |  |  |  |
> | **Qwen3-4B** | Base | 93.00 | 90.80 | **95.80** | **78.80** | 52.02 | **82.90** | 92.62 | 38.49 | **58.70** | **95.20** | **86.16** | 80.00 | 65.60 | **83.50** | 71.50 | 82.40 | 75.76 | 75.90 | 57.48 | **78.44** | **63.02** | 70.02 | **78.11** | 71.82 | 75.82 |
> |  | SFT | 93.60 | 91.20 | 95.20 | 78.00 | **53.27** | 82.06 | **93.47** | 39.28 | 57.20 | 93.88 | 85.31 | 79.68 | **67.70** | 74.40 | 70.73 | 83.06 | 75.85 | **78.31** | 59.26 | 77.55 | 62.74 | 68.66 | 77.45 | 72.02 | 75.47 |
> |  | MAE (half reference) | **94.60** | **92.00** | 95.40 | 77.00 | 49.42 | 82.89 | 92.18 | **39.35** | 58.03 | 94.87 | 84.10 | **81.31** | 65.30 | 76.20 | **78.84** | **85.55** | **76.50** | 78.31 | **62.57** | 81.38 | 62.19 | **70.56** | 77.33 | **74.49** | **76.30** |
>
> **[Comparison with MALT and Novelty Clarification]** Thank you for your question about the novelty of our work. The most significant difference between MALT and MAE is the presence of Proposer and its interactions with the other two roles. As we have clearly shown repeatedly in the original manuscript (**Section 4.1, Figure 1, and Figure 2**), MAE leverages the interactions between every pair of the three agents to enable self-evolution.
>
> To better illustrate the significance of our design, we have demonstrated the **Desirable Difficulty** **Effect** in the revised manuscript (**Section 5.2.2 and Figure 3**), a key factor for self-evolution in MAE. While the rise in generated question difficulty is not always present during training (though it is highly possible), models that exhibit it show **greater and more lasting improvement**. The **Solver-Proposer interaction** is the fundamental source of this effect, since together they iteratively produce more challenging tasks and adapt to them. This indicates that without the Proposer, the model will not yield improvements as significant as ours, and our design’s novelty stands out.

---

> ### Author Response · Authors · 2025-11-25
>
> **[Judge against Ground Truth]** Thank you for raising this key question concerning the design of our framework. However, we argue that Multi-Agent Evolve has the potential to serve as a future training alternative.
>
> We have added experiments with a zero setting, where only **16 self-synthesized seed data** are available. The results demonstrate that the model still achieves similar improvements, thereby showing the advantage of employing an independent judge role rather than using ground truth. Most importantly, these results also illustrate MAE’s potential to operate under **zero prior data**. With MAE, models can gradually improve with minimal human supervision or curation, a promising and **sustainable** approach for LLMs. We believe that such an approach is both necessary and novel.
>
> Given the resources invested in LLM training, we may exhaust all high-quality data one day or even fail to curate data that could contribute to greater intelligence. Leveraging LLM’s own ability for further self-driven improvement allows it to take a further leap forward and not be limited by data curation.
>
> We also argue that using a judge is actually a much more general approach than merely using ground truth, since ground truth does not exist in **open-ended** questions.
>
> **[Collapse Due to Judge]** We have not observed any significant collapse in training due to any role specifically in our refined experiments.
>
> **[Judge Capability]** We would like to kindly point out your misunderstanding regarding our setting and emphasize that activating self-evolution is our primary goal. Therefore, the entire framework of Multi-Agent Evolve **relies solely on the base model** and requires no stronger model to supervise. The use of the LLaMA 70B variant is **limited to evaluation**, and its task is to simply output TRUE or FALSE by comparing the model's answer to the ground truth. Its usage is demonstrated in our revised manuscript (**Appendix B.4**).

---

### Official Review · Reviewer_Mp6f · 2025-11-01

**Soundness:** 2
**Presentation:** 3
**Contribution:** 2
**Rating:** 2
**Confidence:** 3

**Summary:**

This work proposes a multi-agent evolve (MAE) framework with 3 agents: a proposer, a solver, and a judge. The proposer generates questions that the solver attempts to answer and the judge is used to evaluate that response. In tandem, the 3 agents are given a reward for each example based on the responses from the other agent(s). Notably, the rewards are domain-agnostic and therefore the entire process can be run with little to no human supervision. The authors find that doing so leads to improvement of Qwen2.5-3B-Instruct, highlighting the benefit of their data-efficient method.

**Strengths:**

* The paper proposes a framework (MAE) where a Proposer agent generates problem instances (possibly grounded on reference) for a Solver agent; this integration and training of a data generator is a little different from mainstream work which pre-generates the data at once; this likely leads to better data/compute efficiency (although it is not measured).
* Empirical results show that using MAE with some references leads to a Solver which is better than SFT on the seed data.
* Additional analyses highlight the importance of both the iterative refinement ability of the Proposer and the importance of a trained Proposer and Judge.

**Weaknesses:**

The main contribution (of MAE) is not well-situated against simpler baselines. The only baseline compared against is SFT on the seed data (and baseline, no tuning). While half-ref works on held-out sets, it is not a clear winner while with-ref and no-ref perform about the same as without MAE at all. This does not offer a convincing endorsement of using MAE when one would not know when or why to use seed data.

Furthermore, MAE seems to require unnecessary complexity (see below question) over the paradigm of data generation, followed by filtering, followed by solver training.

**Questions:**

1. L143-145 states that MAE is different because the judge is used for evaluation so it is no longer zero-sum. However, some of the papers cited in this paragraph, like Lin et al., 2025, also have a solver and verifier (judge) framework. Is the self-play in this work different from that?

2. Why is zero-sum self-play (L161) a relevant preliminary?

3. Complexity (related to what was mentioned in Weakness): why must the agents co-evolve? The judge is entirely independent of Proposer and Solver. The Judge is only receiving a format reward, so that can be tuned on the seed data (both problems and solutions). Next, the Proposer is only receiving reward from the Judge, so there could be a separate Proposer<>Judge loop with the same reward as mentioned. Finally, the Proposer can either be 1) used to generate a large dataset en masse (with the Judge acting as a filter) or 2) used in the loop with the Solver, in which the Proposer is now fixed. In 1), that dataset could be even used for a new SFT baseline – it is an expanded dataset of “good” questions as proposed by the judge. I suspect this kind of framework already exists, but even if it doesn’t it is a natural and simpler baseline than having the three agents coevolve but the dependencies be arguably acyclic.
	* 5.3.2 doesn’t answer this question because we definitely want to train the Judge and Proposer – my point is why does it need to be done together rather than in 3 separate steps?

4. Why does half-ref work better than no-ref or with-ref?

5. Related to 5.3.3 and 5.3.2: in the no-ref and half-ref setting, why do we even need to train a Solver? The Proposer is solving its own question anyway, so how strong is the Proposer on its own - what additional value does the Solver provide?

---

> ### Author Response · Authors · 2025-11-25
>
> **[Results not Convincing Enough]** We recognize the need to make our claim more convincing. Therefore, we have added new benchmarks and additional experiment settings to demonstrate the effectiveness of Multi-Agent Evolve. The results are also improved with our refined framework designs. Please refer to the general official comment and our revised manuscript for more results.
>
> **[Zero-Sum Self-Play]** The interaction between Solver and Proposer resembles the zero-sum self-play paradigm with some adjustments. Therefore, we still include zero-sum self-play as our preliminary. As we have always stated in our manuscript, we believe that intelligence emerges from interactions, and more complex interactions than self-play are required. Based on this insight, we introduce the Judge into the framework, making it more than just zero-sum self-play since the Proposer and Solver receive rewards **that do not depend on outperforming the other**.
>
> **[Why Coevolve]** Thank you for raising this key question concerning the design of our framework. We would like to kindly point out a critical error in your statement. **Our proposer is not only receiving a reward from the Judge.** It also gets a difficulty reward based on how solvers perform on the questions it generates, which significantly distinguishes us from the pipeline you have proposed. As we have clearly shown repeatedly in the original manuscript (**Section 4.1, Figure 1, and Figure 2**), MAE leverages the interactions between every pair of the three agents to enable self-evolution.
>
> To better illustrate the significance of our design, we have demonstrated the **Desirable Difficulty Effect** in the revised manuscript (**Section 5.2.2 and Figure 3**), a key factor for self-evolution in MAE. While the rise in generated question difficulty is not always present during training (though it is highly possible), models that exhibit it show **greater and more lasting improvement**. The **Solver-Proposer interaction** is the fundamental source of this effect, since together they iteratively produce more challenging tasks and adapt to them. This indicates that separating the Solver and the Proposer during training will not yield improvements similar to ours, and that the co-evolve setting is needed in MAE.
>
> **[Half Reference Outperforms]** The “half reference” and “no reference” settings intuitively have an advantage, as they can generate **original questions** that are beneficial for model improvement, rather than relying solely on **reference questions and their variations**. Experiment results also validate this intuition. We have also provided a more precise explanation of our settings (**Section 5**) for you.
>
> **[Answer Generation]** As mentioned above, the Desirable Difficulty Effect arises from Proposer-Solver interaction. Therefore, it is reasonable to claim that Solver training is always needed.
>
> After identifying the chat template error in the evaluation setup, we found that answer generation is not critical to model performance and has therefore been removed in our new experiments. However, answer generation can still be integrated into the system by assigning a reward for the proposer’s answer to its own question.

---

### Official Review · Reviewer_JnJk · 2025-11-02

**Soundness:** 2
**Presentation:** 3
**Contribution:** 2
**Rating:** 4
**Confidence:** 2

**Summary:**

The paper proposes Multi-Agent Evolve (MAE), a multi-agent reinforcement learning framework that enables large language models (LLMs) to self-improve without human supervision or external verifiers. MAE instantiates three interacting agents (all derived from the same base LLM): Proposer, Solver, and Judge. These agents co-evolve through a synchronized reinforcement learning loop using Task-Relative REINFORCE++. The Judge provides self-reward signals (quality, difficulty, and format rewards) that guide optimization for all three roles. The authors demonstrate consistent improvements over the base model on Qwen2.5-3B-Instruct across multiple benchmarks (math, coding, reasoning, and general knowledge).

**Strengths:**

1. The self-evolving multi-agent framework with no verifiable reward or environment feedback and the synchronized parameter update is novel.

2. The paper is well-written and easy to follow.

**Weaknesses:**

1. There are only vanilla prompting and SFT baselines, no other multi-agent self-play baselines at all. Hard to understand if this is better or worse than self-play with verifiable rewards.

2. Only one base model (Qwen2.5-3B-Instruct) is used. More models are needed to validate the generalizability of the framework.

3. The failure mode of dataset corruption is only briefly mention in section 5.2 without further explanation of why and how this is solved by improving the prompt or applying format rewards.

4. typo: abstract line 27, no percentage number for improvement.

**Questions:**

Why SFT is worse than base (even with a large margin) for quite a few datasets in table 1?

---

> ### Author Response · Authors · 2025-11-25
>
> **[Baseline of Multi-Agent Self-Play]** Thank you for highlighting the need to compare our method with multi-agent self-play baselines. However, we would like to point out that we are among the first frameworks to enable model self-improvement completely via multi-agent interactions. Methods that involve self-play with verifiable rewards are not directly applicable in our setting and may not serve as an informative baseline, since they can operate only in limited domains.
>
> To address your concern and demonstrate the superiority of our approach over self-play methods with verifiable rewards, we have added a comparison with **Absolute Zero Reasoner (AZR)**. Using **Qwen2.5-3B-Instruct** as the backbone, we trained AZR and evaluated it under identical settings with MAE. We also added an experiment with a new setting (zero) for fair comparison, where only 16 self-synthesized seed data are available. As shown in the table below (and added to the revised manuscript), our method outperforms the self-play baseline:
> | Method | GSM8K | MATH | ARC-C | MMLU | GPQA | CQA | OBQA | NQ | TriviaQA | SQuAD | BoolQ | HellaSwag | MBPP+ | HumanEval+ | TruthfulQA | BBH | LiveBench | AMC | Minerva | Winogrande | Olympiad | MMLU-Pro | ID Avg. | OOD Avg. | Overall Avg. |
> | --- | --- | --- | --- | --- | --- | --- | --- | --- | --- | --- | --- | --- | --- | --- | --- | --- | --- | --- | --- | --- | --- | --- | --- | --- | --- |
> | AZR (Self-play) | 81.20 | 62.40 | 82.80 | 63.40 | **38.93** | 74.36 | 77.55 | 35.33 | 56.67 | 90.85 | **78.50** | 67.56 | 61.40 | 68.30 | 44.92 | 52.57 | 19.27 | 34.94 | **41.22** | **64.65** | 28.94 | 44.09 | 67.09 | 41.33 | 57.72 |
> | **MAE (zero)** | **86.00** | **68.20** | **84.20** | 61.40 | 29.42 | 71.54 | 79.39 | **37.98** | 60.13 | 92.28 | 78.46 | 70.34 | **62.20** | **69.50** | **52.71** | **57.51** | 16.48 | **44.58** | 39.91 | 59.48 | 26.47 | 42.66 | 68.37 | 42.48 | 58.51 |
> | **MAE (**half reference**)** | 82.20 | 65.80 | 83.20 | **69.00** | 30.80 | **77.20** | **80.00** | 36.40 | **61.00** | **93.40** | 78.00 | **79.00** | 61.10 | 68.30 | 50.55 | 53.14 | **33.61** | 35.54 | 38.26 | 60.71 | **32.46** | **47.43** | **68.95** | **43.96** | **59.87** |
>
> **Analysis**: MAE (zero) consistently outperforms AZR across all aggregate metrics (Overall Avg. **58.51** vs. 57.72). On standard reasoning tasks, MAE surpasses AZR (e.g., **MATH**: 68.20 vs. 62.40, **+5.80**; **AMC**: 44.58 vs. 34.94, **+9.64**), indicating that our multi-agent evolution strategy pushes the reasoning boundary more effectively than AZR. Furthermore, MAE achieves superior gains over AZR on most commonsense and knowledge benchmarks (e.g., **TruthfulQA**, **HellaSwag**), demonstrating the robustness of our approach across diverse domains.

---

> ### Author Response · Authors · 2025-11-25
>
> **[More Base Models]** We initially focused on the 3B scale due to the high computational cost of RL-based self-play (involving multiple agents and extensive sampling). This constraint is also observed in recent relevant works; for instance, **Kuba et al. (2025, *Language self-play for data-free training*)** similarly validate their method primarily on 3B models.
>
> However, to fully address your concern, we have extended our experiments to **Qwen3-4B-Instruct-2507** and **Qwen2.5-7B-Instruct**. (Due to resource limitations, we were unable to train for long enough or to conduct repeated experiments for each model.)
>
> Comparing our MAE (half reference) against the Base model and SFT baseline, we observed consistent improvements:
>
> - On **Qwen3-4B-Instruct-2507**: MAE (half reference) outperforms the Base model by **0.48** points and SFT by **0.83** points on the overall average. Notably, MAE demonstrates strong generalization, surpassing the Base model by **2.67** points on OOD benchmarks.
> - On **Qwen2.5-7B-Instruct**: MAE (half reference) achieves a clear lead over both Base (**+0.66**) and SFT (**+0.77**), proving that our multi-agent co-evolution framework generalizes effectively to larger model scales.
>
> Detailed results are provided in the table below.
> | Model | Method | GSM8K | MATH | ARC-C | MMLU | GPQA | CQA | OBQA | NQ | TriviaQA | SQuAD | BoolQ | HellaSwag | MBPP+ | HumanEval+ | TruthfulQA | BBH | LiveBench | AMC | Minerva | Winogrande | Olympiad | MMLU-Pro | ID Avg. | OOD Avg. | Overall Avg. |
> | --- | --- | --- | --- | --- | --- | --- | --- | --- | --- | --- | --- | --- | --- | --- | --- | --- | --- | --- | --- | --- | --- | --- | --- | --- | --- | --- |
> | **Qwen2.5-7B** | Base | 91.40 | 74.40 | 91.00 | **74.40** | 30.27 | 81.56 | 87.83 | 42.16 | 67.15 | 96.20 | **87.98** | 81.06 | **68.00** | 74.40 | 63.19 | 69.57 | 27.28 | 49.40 | 53.35 | **68.16** | 37.93 | 55.45 | 74.84 | 53.04 | 66.92 |
> |  | SFT | **92.00** | 74.00 | 91.00 | 74.20 | 30.65 | 81.56 | 88.00 | 41.86 | **67.83** | 95.70 | 87.49 | 81.06 | 67.70 | 74.40 | 63.97 | 70.00 | 26.61 | **50.60** | 49.02 | 68.08 | **38.13** | 56.03 | 74.82 | 52.81 | 66.81 |
> |  | MAE (half reference) | 91.80 | **76.20** | **93.40** | 73.20 | **34.42** | **82.85** | **91.34** | **45.60** | 67.80 | **96.52** | 87.83 | **81.39** | 65.30 | **76.20** | **67.07** | **71.09** | **28.92** | 42.17 | **57.15** | 67.13 | 32.69 | **56.79** | **75.99** | **52.88** | **67.58** |
> |  |  |  |  |  |  |  |  |  |  |  |  |  |  |  |  |  |  |  |  |  |  |  |  |  |  |  |
> | **Qwen3-4B** | Base | 93.00 | 90.80 | **95.80** | **78.80** | 52.02 | **82.90** | 92.62 | 38.49 | **58.70** | **95.20** | **86.16** | 80.00 | 65.60 | **83.50** | 71.50 | 82.40 | 75.76 | 75.90 | 57.48 | **78.44** | **63.02** | 70.02 | **78.11** | 71.82 | 75.82 |
> |  | SFT | 93.60 | 91.20 | 95.20 | 78.00 | **53.27** | 82.06 | **93.47** | 39.28 | 57.20 | 93.88 | 85.31 | 79.68 | **67.70** | 74.40 | 70.73 | 83.06 | 75.85 | **78.31** | 59.26 | 77.55 | 62.74 | 68.66 | 77.45 | 72.02 | 75.47 |
> |  | MAE (half reference) | **94.60** | **92.00** | 95.40 | 77.00 | 49.42 | 82.89 | 92.18 | **39.35** | 58.03 | 94.87 | 84.10 | **81.31** | 65.30 | 76.20 | **78.84** | **85.55** | **76.50** | 78.31 | **62.57** | 81.38 | 62.19 | **70.56** | 77.33 | **74.49** | **76.30** |
>
> **[Explanation of Datasets Corruption and Prevention]** Thank you for raising this important point. We have significantly expanded our explanation of dataset corruption in the revised manuscript (**Section 5.2.2** and **Figure 4**).
>
> Dataset corruption refers to the deterioration in a dataset's quality over time and is now resolved by using a stricter data quality filter. Since then, our training process has demonstrated greater stability than previous pipelines. Through our experiment, we found that refining the prompt and using a format reward also helps alleviate dataset corruption. However, their effect is not strong enough to be comparable to a strict data quality filter, which directly prevents such issues from occurring.
>
> We have added new ablation experiments on question-quality filtering and the format reward in the revised manuscript.  The results show that applying a strict question-quality filtering is the key to preventing dataset corruption and improving stability.
>
> **[Writing Issues]** We have corrected the typo in the revised abstract. The text now explicitly states that MAE achieves an average improvement of 4.86% across multiple benchmarks, reflecting the results from our updated experiments.

---

> ### Author Response · Authors · 2025-11-25
>
> **[SFT Baseline Performance]** We examined the SFT baseline and identified a **chat template mismatch** (between training and evaluation) in our initial setup. After correcting this and re-running the experiments, the updated SFT performance and corresponding analysis are shown in our revised manuscript.

---

### Author Response · Authors · 2025-11-25

We thank the reviewers for their time. We have also been investing significant time and effort in improving Multi-Agent Evolve. We are pleased to have uploaded a revised version of the paper with substantial updates to the framework, experiments, and analysis. The key changes are summarized below:

1. **Refined Settings & Baselines:** We formalized four experimental settings, including a "Zero" setting (starting with only 16 self-synthesized seeds) to fairly compare against a self-play RL baseline (with a verifiable environment), **Absolute Zero Reasoner (AZR)**.
2. **Evaluation Update and Expanded Evaluation:** We identified a chat template error in our initial evaluation setup that caused all performance evaluations to be lower than expected. We also introduced more benchmarks and categorized them into **In-Distribution** (e.g., SQUAD, ARC-C; **Same Distribution with Seed Data**) and **Out-Of-Distribution** (e.g., Minerva, MMLU-Pro; **Held-out Datasets**) to rigorously test the model’s performance across diverse domains. More experiments are carried out under different settings as well.
3. **Methodology Update:** We simplified the Proposer by removing the "answer-then-refine" requirement. Instead, we addressed dataset corruption and ensured stability through **Quality Filtering** (threshold > 0.7) and Format Rewards.
4. **New Analysis:** We identify a **"Desirable Difficulty Effect”**, demonstrating that model performance gains correlate positively with the rising difficulty of generated questions during training. This effect illustrates the necessity of our multi-agent setting.

Below, we summarize the key points reviewers raised—items marked with ** indicate issues for which we provide additional experiments or clarifications, while unmarked items reflect strengths they acknowledged. "Action/Summary" includes the highly summarized rebuttal content for each reviewer.
| **Dimension** | **Reviewer JnJk** | **Reviewer Mp6f** | **Reviewer dmfX** | **Reviewer Pegt** | **Action / Summary** |
| --- | --- | --- | --- | --- | --- |
| **1. Novelty & Significance** | “Novel multi-agent self-evolving framework without human supervision.” | “Integration of Proposer–Solver–Judge is distinct from standard self-play pipelines.” | **“Can authors clarify … MALT?”, ”For such questions, … Why not use them?”** | “The idea of using three distinct roles (Proposer, Solver, Judge) co-evolving … beyond zero-sum games or verifiable environments.” | **Summary:** 3 out of 4 reviewers recognize the timeliness and novelty of Multi-Agent Evolve. **Reviewer** **dmfX rebuttal:** We conduct experiment on MAE (zero) that relies on no prior data. Results show similar improvement which demonstrate MAE’s novelty compared to MALT and potential for future training. Besides, the Desirable Difficulty Effect also shows the significance of Proposer-Solver interaction, further distinguishing MAE from MALT. |
| **2. Methodology & Validity** | N/A | **“why must the agents co-evolve? … but the dependencies be arguably acyclic.”** | N/A | N/A | **Rebuttal:** 3 out of 4 reviewers acknowledge methodological validity. **Reviewer Mp6f rebuttal:** We added training process analysis which identifies an important Desirable Difficulty Effect. This effect emerges from Proposer-Solver interaction and leads to lasting performance in training.  |
| **3. Evaluation Scope and Base Models** | **“More models are needed to validate the generalizability of the framework.”** | N/A | **“The main concern is that the framework is evaluated on a single backbone LLM of relatively small size (3B parameters).”** | N/A | **Summary:** We added more benchmarks for evaluation. We also providde results on Qwen2.5-7B-Instruct and Qwen3-4B-Instruct-2507. MAE results consistently outperforms base model and SFT baseline.  |
| **4. Presentation** | “Well-written and organized.” | “Readable and coherent.” | “Figures clear and informative.” | **“Need clearer expressions for reward functions of different roles.”** | **Summary:** 3 out of 4 reviewers praise clarity. Minor typo issues are corrected and we have clarified our settings further. New appendix added for reproducibility details and training scripts. We also provide our code to the reviewers. |

---

### Author Response · Authors · 2025-11-25

**We conducted additional experiments on larger and newer models: Qwen3-4B-Instruct-2507 and Qwen2.5-7B-Instruct with “half reference” setting.** (Due to resource limitations, we were unable to train for long enough or to conduct repeated experiments for each model.)

Comparing our method (MAE) against the Base model and SFT baseline, we observed consistent improvements:

- On **Qwen3-4B-Instruct-2507**: MAE (half reference) outperforms the Base model by **0.48** points and SFT by **0.83** points on the overall average. Notably, MAE (half reference) demonstrates significant generalization capability, surpassing the Base model by **2.67** points on OOD benchmarks.
- On **Qwen2.5-7B-Instruct**: MAE (half reference) achieves a clear lead over both Base (**+0.66**) and SFT (**+0.77**), proving that our multi-agent co-evolution framework generalizes effectively to larger model scales.

Detailed results are provided in the table below.
| Model | Method | GSM8K | MATH | ARC-C | MMLU | GPQA | CQA | OBQA | NQ | TriviaQA | SQuAD | BoolQ | HellaSwag | MBPP+ | HumanEval+ | TruthfulQA | BBH | LiveBench | AMC | Minerva | Winogrande | Olympiad | MMLU-Pro | ID Avg. | OOD Avg. | Overall Avg. |
| --- | --- | --- | --- | --- | --- | --- | --- | --- | --- | --- | --- | --- | --- | --- | --- | --- | --- | --- | --- | --- | --- | --- | --- | --- | --- | --- |
| **Qwen2.5-7B** | Base | 91.40 | 74.40 | 91.00 | **74.40** | 30.27 | 81.56 | 87.83 | 42.16 | 67.15 | 96.20 | **87.98** | 81.06 | **68.00** | 74.40 | 63.19 | 69.57 | 27.28 | 49.40 | 53.35 | **68.16** | 37.93 | 55.45 | 74.84 | 53.04 | 66.92 |
|  | SFT | **92.00** | 74.00 | 91.00 | 74.20 | 30.65 | 81.56 | 88.00 | 41.86 | **67.83** | 95.70 | 87.49 | 81.06 | 67.70 | 74.40 | 63.97 | 70.00 | 26.61 | **50.60** | 49.02 | 68.08 | **38.13** | 56.03 | 74.82 | 52.81 | 66.81 |
|  | MAE (half reference) | 91.80 | **76.20** | **93.40** | 73.20 | **34.42** | **82.85** | **91.34** | **45.60** | 67.80 | **96.52** | 87.83 | **81.39** | 65.30 | **76.20** | **67.07** | **71.09** | **28.92** | 42.17 | **57.15** | 67.13 | 32.69 | **56.79** | **75.99** | **52.88** | **67.58** |
|  |  |  |  |  |  |  |  |  |  |  |  |  |  |  |  |  |  |  |  |  |  |  |  |  |  |  |
| **Qwen3-4B** | Base | 93.00 | 90.80 | **95.80** | **78.80** | 52.02 | **82.90** | 92.62 | 38.49 | **58.70** | **95.20** | **86.16** | 80.00 | 65.60 | **83.50** | 71.50 | 82.40 | 75.76 | 75.90 | 57.48 | **78.44** | **63.02** | 70.02 | **78.11** | 71.82 | 75.82 |
|  | SFT | 93.60 | 91.20 | 95.20 | 78.00 | **53.27** | 82.06 | **93.47** | 39.28 | 57.20 | 93.88 | 85.31 | 79.68 | **67.70** | 74.40 | 70.73 | 83.06 | 75.85 | **78.31** | 59.26 | 77.55 | 62.74 | 68.66 | 77.45 | 72.02 | 75.47 |
|  | MAE (half reference) | **94.60** | **92.00** | 95.40 | 77.00 | 49.42 | 82.89 | 92.18 | **39.35** | 58.03 | 94.87 | 84.10 | **81.31** | 65.30 | 76.20 | **78.84** | **85.55** | **76.50** | 78.31 | **62.57** | 81.38 | 62.19 | **70.56** | 77.33 | **74.49** | **76.30** |

We would also like to emphasize our motivation for this work: **model self-evolution**. Therefore, our designs focus on improvements **without supervision or verification**. Our results are **completely reproducible**, and we provide our code for reviewers at https://anonymous.4open.science/r/Multi-Agent-Evolve-Anonymous-4558.

---

### Meta-Review · Area_Chair_r3Td · 2026-01-06

**Summary:**

This paper proposes a multi-agent reinforcement learning framework that enables large language models (LLMs) to self-improve.  The reviewers raised the following concerns:

1. The technical novelty of the contribution appears limited, and the distinctions from prior work need to be more clearly articulated.

2. The proposed approach is relatively complex, yet it does not demonstrate significant improvements over simple baselines.

3. The experiment section is weak.

Although the proposed approach is promising, the paper is not recommended for acceptance in its current form. The authors are encouraged to address the reviewers’ feedback and further strengthen the work for resubmission to other venues.

**Reviewer Concerns:**

The reviewer conducted additional experiments to evaluate the effectiveness of the approach; however, these results were not sufficient to persuade reviewers to shift to a positive assessment.

**Reviewer Scores:**

See above.

---

### Decision · Program_Chairs · 2026-01-26

Reject